

**Sensitivity of cloud phase distribution to cloud microphysics and**
**thermodynamics in simulated deep convective clouds and SEVIRI**
**retrievals**
Cunbo Han[1,2], Corinna Hoose[1], Martin Stengel[3], Quentin Coopman[4], Andrew Barrett[1]
1. Institute of Meteorology and Climate Research (IMK-TRO), Karlsruhe Institute of
Technology, Karlsruhe, Germany
2. Now at State Key Laboratory of Tibetan Plateau Earth System, Environment and
Resources (TPESER), Institute of Tibetan Plateau Research, Chinese Academy
of Sciences, Beijing, China
3. Deutscher Wetterdienst (DWD), Offenbach, Germany
4. Department of Atmospheric and Oceanic Sciences, McGill University, Montreal,
Canada
Correspondence to: Cunbo Han (cunbo.han@hotmail.com) and Corinna Hoose
(corinna.hoose@kit.edu)



**Abstract:**
The formation of ice in clouds is an important process in mixed-phase clouds, and
the radiative properties and dynamical developments of clouds strongly depend on
their partitioning between liquid and ice phases. In this study, we investigate the
sensitivities of the cloud phase to ice-nucleating particle (INP) concentration and
thermodynamics. Experiments are conducted using the ICOsahedral Nonhydrostatic
model (ICON) at the convection-permitting resolution of about 1.2 km on a domain
covering significant parts of central Europe, and are compared to two different
retrieval products based on SEVIRI measurements. We select a day with several
isolated deep convective clouds, reaching a homogeneous freezing temperature at
the cloud top. The simulated cloud liquid pixel number fractions are found to
decrease with increasing INP concentration both within clouds and at the cloud top.
The decrease in cloud liquid pixel number fraction is not monotonic but is stronger in
high INP cases. Cloud-top glaciation temperatures shift toward warmer temperatures
with increasing INP concentration by as much as 8 °C. Moreover, the impact of INP
concentration on cloud phase partitioning is more pronounced at the cloud top than
within the cloud. Moreover, initial and lateral boundary temperature fields are
perturbed with increasing and decreasing temperature increments from 0 to +/-3K
and +/-5K between 3 and 12 km. Perturbing the initial thermodynamic state is also
found to affect the cloud phase distribution systematically. However, the simulated
cloud-top liquid number fraction, diagnosed using radiative transfer simulations as
input to a satellite forward operator and two different satellite remote sensing
retrieval algorithms, deviates from one of the satellite products regardless of
perturbations in the INP concentration or the initial thermodynamic state for warmer
sub-zero temperatures, while agreeing with the other retrieval scheme much better,
in particular for the high INP and high convective available potential energy (CAPE)
scenarios. Perturbing the initial thermodynamic state, which artificially increases the
instability of the mid- and upper-troposphere, brings the simulated cloud-top liquid
number fraction closer to the satellite observations, especially in the warmer mixed-
phase temperature range.

**Keywords**: Mixed-phase clouds, deep convection, INP, thermodynamics, satellite
forward operator, remote-sensing retrieval algorithms






**Key points:**
1. Cloud properties are retrieved using a satellite forward operator and remote
sensing retrieval algorithms with ICON simulations as input. To our knowledge,
it is the first time this approach has been used to retrieve cloud phase and other
microphysical variables.
2. Glaciation temperature shifts towards a warmer temperature with increasing
INP concentration both within the cloud and at the cloud top. Initial
thermodynamic states affect the cloud phase distribution significantly as well.
3. Simulated cloud-top pixel number fraction matches the satellite observations in
the high INP and high CAPE scenarios.




## 1. Introduction


In the temperature range between 0 and -38°C, ice particles and supercooled liquid
droplets can coexist in mixed-phase clouds. Mixed-phase clouds are ubiquitous in
Earth's atmosphere, occurring at all latitudes from the poles to the tropics. Because
of their widespread nature, mixed-phase processes play a critical role in the life cycle
of clouds, precipitation formation, cloud electrification, and the radiative energy
balance on both regional and global scales (Korolev et al., 2017). Deep convective
clouds are always mixed-phase clouds, and their cloud tops reach the homogeneous
freezing temperature, -38°C, in most cases. Despite the importance of mixed-phase
clouds in shaping global weather and climate, microphysical processes for mixed-
phase cloud formation and development are still poorly understood, especially ice
formation processes. It is not surprising that the representation of mixed-phase
clouds is one of the big challenges in weather and climate models (McCoy et al.,
2016; Korolev et al., 2017; Hoose et al., 2018; Takeishi and Storelvmo, 2018; Vignon
et al., 2021; Zhao et al., 2021).

The distribution of cloud phase has been found to impact cloud thermodynamics and
Earth's radiation budget significantly (Korolev et al., 2017; Matus and L'Ecuyer,
2017; Hawker et al., 2021). The freezing of liquid droplets releases latent heat and
hence affects the thermodynamic state of clouds. Moreover, distinct optical
properties of liquid droplets and ice particles exert different impacts on cloud's
shortwave and longwave radiation. Observational studies reveal that the cloud phase
distribution is highly temperature-dependent and influenced by multiple factors, for
example, cloud type and cloud microphysics (Rosenfeld et al., 2011; Coopman et al.,
2020). Analyzing passive satellite observations of mixed-phase clouds over the
Southern Ocean, Coopman et al. (2021) found that cloud ice fraction increases with
increasing cloud effective radius. Analysis of both passive and active satellite
datasets reveals an increase in supercooled liquid fraction with cloud optical
thickness (Bruno et al., 2021).

A number of in-situ observations of mixed-phase clouds have been made in the past
several decades, covering stratiform clouds (Pinto, 1998; Korolev and Isaac, 2006;
Noh et al., 2013) and convective clouds (Rosenfeld and Woodley, 2000; Stith et al.,



2004; Taylor et al., 2016). Aircraft-based observations of mixed-phase clouds
properties reveal that the frequency distribution of the ice water fraction has a U-
shape, with the occurrence of mixed-phase clouds decreasing toward lower
temperatures (Korolev et al., 2003; Field et al., 2004; Korolev et al., 2017). These
findings are very useful constraints of numerical models (Lohmann and Hoose, 2009;
Grabowski et al., 2019). However, in-situ observations of mixed-phase cloud
microphysics are technically difficult and sparse in terms of spatial and temporal
coverage. Thus, understanding ice formation processes and determining the
climatological significance of mixed-phase clouds have proved difficult using existing
in-situ observations only.

Both observations and simulations reveal that INPs impact deep convective cloud
properties including the persistence of deep convective clouds and precipitation
(Twohy, 2015; Fan et al., 2016). Satellite observations indicate that dust serves as
effective INPs in the Saharan air layer, promotes the heterogeneous ice nucleation
process, shifts the precipitation size distribution from large to small raindrops in deep
convective clouds, and ultimately reduces precipitation (Min et al., 2009). However,
the convection-permitting simulations by van den Heever et al. (2006) showed that
convective precipitation increases with increasing INPs. Moreover, some simulation
studies argue that dust aerosols acting as INPs have hardly any effect on convective
precipitation although they significantly impact cloud microphysical properties (Fan et
al., 2010; Fan et al., 2016). Li and Min (2010) suggested that the impacts of INPs on
deep convective precipitation systems highly depend on the precipitation type.
Although the effects of INPs on convective precipitation are not conclusive, it is
certain that the interactions between convective clouds and INPs affect cloud
microphysical properties and hence cloud phase distributions. In addition, previous
numerical modeling studies on cloud-aerosols interactions have focused on
influences of aerosols acting as cloud condensation nuclei (CCN) (Fan et al., 2016),
which are linked to the ice phase e.g. through impacts on the riming efficiency
(Barrett and Hoose, 2023). Given the limited knowledge on ice formation in deep
convective clouds and significant uncertainties in ice nucleation parameterizations, it
is necessary to conduct sensitivity simulations to investigate how ice formation
processes are influenced by INP concentrations and thermodynamic states in deep
convective clouds.




In this study, with the help of realistic convection-permitting simulations using two-
moment microphysics, we address how and to what extent INP concentration and
thermodynamic state affect the in-cloud and cloud-top phase distributions in deep
convective clouds. In particular, cloud properties are retrieved using a satellite
forward operator and remote sensing retrieval algorithms with radiative transfer
simulations as input for a fair comparison to observations from SEVIRI. A similar
strategy was used by Kay et al. (2018) for the evaluation of precipitation in a climate
model with CloudSat observations and termed "scale-aware and definition-aware
evaluation". Stengel et al. (2020) applied a cloud classification algorithm developed
for satellite observations to model simulated brightness temperatures in a similar
manner. This method allows us to compare model simulated cloud properties with
remote sensing cloud products directly, and is, to our knowledge, the first time this
approach is used for the cloud phase and related microphysical variables.

This paper is structured as follows: In section 2, we introduce our model setups and
the experiment design, the satellite forward operator, remote sensing retrieval
algorithms, and datasets. Simulation results for the sensitivity experiments are
shown in section 3. Section 4 presents discussions; and we summarize the study
and draw conclusions in section 5.
**2.  Data and Method**
**2.1.  Model description**
The Icosahedral Nonhydrostatic (ICON) model (Zängl et al., 2015) is a state-of-the-
art unified modeling system offering three physics packages, which are dedicated to
numerical weather prediction (NWP), climate simulation, and large-eddy simulation.
ICON is a fully compressible model and has been developed collaboratively between
the German Weather Service (DWD), Max Planck Institute for Meteorology, German
Climate Computing Center (DKRZ), and Karlsruhe Institute of Technology (KIT). In
order to maximize the model performance and to remove the singularity at the poles,
ICON solves the prognostic variables suggested by Gassmann and Herzog (2008),
on an unstructured triangular grid with C-type staggering based on a successive
refinement of a spherical icosahedron (Wan et al., 2013). Governing equations are



described in Wan et al. (2013) and Zängl et al. (2015). The DWD has operated the
ICON model at a spatial resolution of about 13 km on the global scale since January
2015. In the global ICON, the higher-resolution ICON-EU (resolution 7 km) nesting
area for Europe has been embedded since July 2015. In this study, ICON-2.6.4 with
the NWP physics package is used and initial and lateral boundary conditions are
provided by the ICON-EU analyses.

For cloud microphysics, we use an updated version of the two-moment cloud
microphysics scheme developed by Seifert and Beheng (2006). The two-moment
scheme predicts the number and mass mixing ratios of two liquid (cloud and rain)
and four solid (ice, graupel, snow, and hail) hydrometers. The cloud condensation
nuclei (CCN) activation is described following the parameterization developed by
Hande et al. (2016). Homogeneous freezing, including freezing of liquid water
droplets and liquid aerosols, is parametrized according to Kärcher et al. (2006).
Heterogeneous ice nucleation, including the immersion and deposition modes, is
parameterized as a function of temperature- and ice supersaturation-dependent INP
concentration (Hande et al., 2015). The INP concentration due to immersion
nucleation is described as the following equation:
$$C_{INP}(T_K) = A \times \exp[-B \times (T_K - T_{min})^C] \qquad (1)$$

where $T_k$ is the ambient temperature in Kelvin; $A$, $B$, and $C$ are fitting constants with
different values to represent seasonally varying dust INP concentrations. The
parameterization for deposition INPs is simply scaled to the diagnosed relative
humidity with respect to ice ($RH_{ice}$):
$$C_{INP}(T_K, RH_{ice}) \approx C_{INP}(T_K) \times DSF(RH_{ice}) \qquad (2)$$

$$DSF(RH_{ice}) = a \times \arctan(b \times (RH_{ice} - 100) + c) + d \qquad (3)$$

where $C_{INP}(T_K)$ is given by Equation (1); $a$, $b$, c, and $d$ are constants. More details
are found in Hande et al. (2015).
**2.2.  Simulation setup and sensitivity experiments**
In this study, the setup consists of two different domains with one-way nesting
covering a major part of central Europe (Figure 1). The horizontal resolution for the
nested domains is halved from 2400 m to 1200 m in the innermost domain, and the
time steps for the two domains are 12 s and 6 s, respectively. 150 vertical levels are



used, with a grid stretching towards the model top at 21 km. The vertical resolution is
the same for all horizontal resolutions and the lowest 1000 m encompass 20 layers.
A 1-D vertical turbulence diffusion and transfer scheme is used for the 2400 m and
1200 m resolutions, referred to as numerical weather prediction (NWP) physics.
Deep convection is assumed to be explicitly resolved, while shallow convection is
parameterized for both domains. The simulations are initialized at 00:00 UTC on the
study day from ICON-EU analyses and integrated for 24 hours. At the lateral
boundaries of the outer domain, the simulation of the model is updated with 3-hourly
ICON-EU analyses. The nested domains are coupled online, and the outer domain
provides lateral boundary conditions to the inner domain.

In nature, INP concentration varies across multiple orders of magnitude (Hoose and
Möhler, 2012; Kanji et al., 2017). Thus, in our sensitivity experiments, heterogeneous
ice formation was scaled by multiplying the default INP concentration (Equation (1))
with a factor of $10^{-2}$, $10^{-1}$, $10^1$, $10^2$, $10^3$ for both immersion freezing and deposition ice
nucleation. Together with a case with default INP concentration (case CTRL) and
one case switching off the secondary-ice production via rime-splintering process (the
so called Hallet-Mossop process), 7 cases were created in total to investigate the
impact of primary and secondary ice formation on cloud phase distribution in deep
convective clouds.

In order to assess the sensitivity of the cloud phase to thermodynamics, initial and
lateral boundary temperature fields are modified with increasing and decreasing
temperature increments, named experiments INC and DEC, respectively. The
temperature increments are linearly increasing/decreasing from 0 to +/-3K and +/-5K
between 3 and 12 km, creating 4 sensitivity experiments DEC03, DEC05, INC03,
and INC05. Above 12 km, the increment is constant up to the model top. Initial
temperature profiles are shown in Figure 2. The increasing or decreasing
environmental temperature leads to changes in the lapse rate and the stability of the
atmosphere, and hence results in decrease or increase in the convective available
potential energy (CAPE), respectively (Barthlott and Hoose, 2018). Thus, the CAPE
increases monotonically from case INC05 (spatial-averaged CAPE at 9:00 UTC: 413
J kg$^{-1}$) to case CTRL (724 J kg$^{-1}$) and finally to DEC05 (1235 J kg$^{-1}$). Note that the
relative humidity increases/decreases with decreasing/increasing temperature as the





specific humidity is unperturbed. The perturbations of INP concentration and
initial/lateral temperature profiles are motivated by Hoose et al. (2018) and Barthlott
and Hoose (2018), respectively. Complementary to these earlier studies, we now
investigate an ensemble of several deep convective clouds and focus on influences
of INP and thermodynamics on cloud phase distribution. Short descriptions of all
sensitivity experiments performed in this study are listed in Table 1.
**2.3.  Satellite observations and retrieval algorithms**
The Spinning Enhanced Visible and Infrared Imager (SEVIRI) is a 12-channel imager
on board the geostationary Meteosat Second Generation (MSG) satellites. SEVIRI
has one high spatial resolution visible channel (HRV) and 11 spectral channels from
0.6 to 14 μm with a 15 min revisit cycle and a spatial resolution of 3 km at nadir
(Schmetz et al., 2002). Based on the spectral measurements of SEVIRI, a cloud
property data record, the CLAAS-2 dataset (CLoud property dAtAset using SEVIRI,
Edition 2), has been generated in the framework of the EUMETSAT Satellite
Application Facility on Climate Monitoring (CM SAF) (Benas et al., 2017). CLAAS-2
is the successor of CLAAS-1 (Stengel et al., 2014), for which retrieval updates have
been implemented in the algorithm for the detection of clouds compared to CLAAS-1
(Benas et al., 2017) with the temporal coverage being extended to 2004-2015.
Retrieval algorithms for parameters that are important for this study are introduced
below. Detailed descriptions for the retrieval algorithms are found in Stengel et al.
(2014) and Benas et al. (2017) with the main features being summarized in the
following.

The MSGv2012 software package is employed to detect clouds and their vertical
placement (Derrien and Le Gléau, 2005; Benas et al., 2017). Multi-spectral threshold
tests, which depend on illumination and surface types, among other factors, are
performed to detect cloud appearances. Each satellite pixel is assigned to categories
of cloud-filled, cloud-free, cloud water contaminated, or snow/ice contaminated.
Cloud top pressure (CTP) is retrieved with different approaches using input from
SEVIRI channels at 6.2, 7.3, 10.8, 12.0, and 13.4 μm (Menzel et al., 1983; Schmetz
et al., 1993; Stengel et al., 2014; Benas et al., 2017). Cloud top height (CTH) and
cloud top temperature (CTT) are derived from CTP using ancillary data for



temperature and humidity profiles from ERA-Interim (Dee et al., 2011). The cloud top
phase (CPH) retrieval is based on a revised version of the multispectral algorithm
developed by Pavolonis et al. (2005). Clouds are categorized initially into six types,
that are liquid, supercooled, opaque ice, cirrus, overlap, and overshooting.
Subsequently, the binary cloud phase (liquid or ice) is generated based on the six
categories (Benas et al., 2017). Cloud optical and microphysical properties are
retrieved using the Cloud Physical Properties (CPP) algorithm (Roebeling et al.,
2006). SEVIRI visible (0.6 μm) and near-infrared (1.6 μm) measurements are used
to calculate cloud optical thickness (COT) and cloud particle effective radius ($r_e$) by
applying the Nakajima and King (1990) approach in the CPP algorithm (Stengel et
al., 2014; Benas et al., 2017). Liquid water path (LWP) and ice water path (IWP) are
then computed as a function of liquid/ice water density, COT, and $r_e$ of cloud water
and cloud ice following the scheme developed by Stephens (1978).

In this study we used instantaneous CLAAS-2 data with temporal resolution of 15
minutes and on native SEVIRI projection and resolution. In addition to the CLAAS-2
dataset, the recently developed software suite SEVIRI_ML (Philipp and Stengel, to
be submitted) was applied to the SEVIRI measurements to obtain cloud top phase
and cloud top temperature for the selected case. SEVIRI_ML uses a machine
learning approach calibrated against CALIOP. One feature of the SEVIRI_ML is that
it also provides pixel-based uncertainties such that values with low reliability can be
filtered out.
**2.4. Satellite forward operators**
In order to compare simulation results and satellite observations directly, SEVIRI-like
spectral reflectance and brightness temperatures are calculated using the radiative
transfer model for TOVS (RTTOV, v12.3)(Saunders et al., 2018). RTTOV is a fast
radiative transfer model for simulating top-of-atmosphere radiances from passive
visible, infrared, and microwave downward-viewing satellite radiometers. It has been
widely used in simulating synthetic satellite images and assimilating radiances in
numerical models (Saunders et al., 2018; Pscheidt et al., 2019; Senf et al., 2020;
Geiss et al., 2021; Rybka et al., 2021).



In this work, ICON simulated surface skin temperature, near-surface pressure,
temperature, specific humidity, wind velocity, total liquid water content, total ice water
content, and effective radius of cloud liquid and cloud ice are used as input to drive
the RTTOV model. Before inputting to the RTTOV model, ICON simulations are
remapped onto SEVIRI's full disc coordinate. Brightness temperatures from 8
channels (at 3.9, 6.2, 7.3, 8.7, 9.7, 10.8, 12.0, and 13.4 $\mu$m) and reflectance from 3
channels (at 0.6, 0.8, and 1.6 $\mu$m) simulated by the RTTOV model are used as input
to run the remote sensing retrieval algorithms to derive CLAAS-2-like and
SEVIRI_ML-like retrievals, named ICON_RTTOV_CLAAS-2 and
ICON_RTTOV_SEVIRI_ML products, respectively.
**2.5. Synoptic overview**
The day 06 June 2016 was selected to analyze, which was dominated by
summertime deep convection located in central Europe. The synoptic forcing was
weak on the day, and convection was triggered mainly by local thermal instabilities.
The day has been discussed frequently in previous studies in terms of convection
triggering, cloud microphysics, and its parameterizations (Keil et al., 2019; Geiss et
al., 2021).
**3. Results and discussion**
Perturbing INP concentration and temperature profiles directly affects microphysical
and thermodynamic processes of the developing deep convective clouds, and hence
impact in-cloud and cloud-top phase distributions. The following section shows
results and discussions on the sensitivities of cloud phase and cloud microphysics to
INP concentration and thermodynamic perturbations.
**3.1. Spatial distribution of cloud properties**
Before analyzing the results of sensitivity experiments, retrieved cloud properties via
RTTOV and the CLAAS-2 retrieval scheme for the CTRL case are compared to
CLAAS-2 products. Spatial distributions of derived LWP, IWP, and COT at 13:00
UTC of the CTRL case and CLAAS-2 satellite observation are shown in Figure *3*.
Discrepancies are found between ICON simulation and CLAAS-2 satellite
observations in terms of spatial coverage and intensity. The ICON simulation



overestimates the cloud coverage of low-level liquid clouds compared to CLAAS-2
satellite observations, while LWP derived from the ICON simulation (case CTRL) is
smaller and more homogeneously distributed than that from the CLAAS-2
observation (Figure 3a and 3b). The spatial distributions of IWP and COT represent
the approximate location and spatial extension of deep convective clouds in this
study. The ICON simulation could reproduce cores of deep convective clouds of a
number and spacing comparable to observations, while the spatial extension and
intensity of individual deep convective clouds are not simulated very well by the
ICON model. The ICON simulation underestimates the spatial extension of deep
convective clouds but overestimates IWP and COT outside the convective cores
compared to the CLAAS-2 observation (Figure *3*c-f). Overall, the simulated clouds
appear to be too homogeneous without sufficient internal structure. Geiss et al.
(2021) also reported significant deviations between model simulations and satellite
observations. Moreover, Geiss et al. (2021) concluded that the primary source of
deviations is mainly from model physics, especially model assumptions on subgrid-
scale clouds.
**3.2.   Sensitivity of microphysical properties to INP perturbation**
Perturbing INP concentration results in a direct influence on the heterogeneous
freezing processes and hence impacts on cloud microphysical properties.
Systematic variations have been found in the spatial- and time-averaged profiles of
mass mixing ratios of cloud hydrometeors as shown in Figure 4. All profiles
discussed here are averaged over cloudy pixels (defined as having a condensed
mass of cloud water plus cloud total cloud ice greater than a threshold of $1.0\times10^{-5}$ kg
$kg^{-1}$) and over the time period from 9:00 to 19:00 UTC, when convection was well
developed. The mass concentration of ice crystals decreases with increasing INP
concentration (Figure 4a). However, the mass concentration of snow, graupel, and
rainwater increase with increasing INP concentration, especially in the high INP
concentration cases (cases $A\times10^2$ and $A\times10^3$).

In order to further reveal why ice crystal mass concentration decreases with
increasing INP concentration, we investigate process rates related to ice particle
nucleation and growth. Figure 5 shows spatial- and time-averaged (from 9:00 to



19:00 UTC) profiles of process rates for homogeneous freezing, heterogeneous
freezing, secondary ice production via the rime-splintering process, cloud droplets
rimed with ice crystals, rain droplets rimed with ice crystals, and collection between
ice and ice crystals. Heterogeneous freezing (Figure 5a) includes processes of
immersion freezing, deposition ice nucleation, and immersion freezing of liquid
aerosols (Kärcher et al., 2006; Hande et al., 2015), see also equations (1) and (2).
Process rates of heterogeneous freezing increase significantly with increasing INP
concentration compared to the CTRL (Figure 5a). Compensating the change in
heterogeneous freezing, process rates of homogeneous freezing decrease
significantly with increasing INP concentration (Figure 5b). However, a decrease in
INP concentration (compared to the CTRL) does not have a strong influence on the
heterogeneous freezing mass rate, which is already low compared to the other
processes in CTRL. Riming processes of cloud droplets and rain droplets onto ice
crystals are greatly invigorated due to enhanced INP concentration (Figure 5d and
5e). Moreover, process rates of secondary ice production due to rime-splintering are
strengthened as well due to the increase in rimed ice, albeit much lower values.
Figure 5f shows process rates of collection between ice and ice crystals. Process
rates of collection between ice and ice particles increase with increasing INP
concentration, especially in high INP concentration cases (cases $A\times10^2$ and $A\times10^3$).
Process rates of collection of other ice particles all increase with increasing INP
concentration, similar to the collection between ice and ice crystals (not shown). The
increase in the riming of clouds and rain droplets onto ice crystals and collections
between ice particles leads to the increase in the mass concentration of snow,
graupel, and hail (Figure 4b and 4c). However, the total mass increase in snow,
graupel, and hail do not outbalance the decrease in the mass concentration of ice
crystals (Figure 4). The weakened homogeneous freezing is most likely the dominant
factor leading to the decrease in ice mass concentration in high INP cases,
considering the magnitude of the process rate of homogeneous freezing (Figure 5b).
Supercooled liquid and cloud droplets have been converted into ice crystals before
reaching the homogeneous freezing layer, leading to fewer supercooled droplets
remaining for homogeneous freezing. Even though homogeneous freezing is
weakened in high INP cases, the process rate of homogeneous freezing is still larger
than heterogeneous freezing, which means homogeneous freezing is the dominant



ice formation process in the convective clouds discussed in this study. Moreover, the
enhanced production of large ice particles (snow, graupel, and hail) in the highest
INP case, which sediment more rapidly to lower levels, leads to increased surface
precipitation by about 10% in the A×10³ case (not shown). Interestingly, ice crystal
effective radius ($r_e^{ice}$) increases monotonically with increasing INP concentration,
especially in the mixed-phase layer (Figure *4*e). Zhao et al. (2019) also reported an
increased $r_e^{ice}$ with polluted continental aerosols in their simulated moderate
convection cases, and they attributed it to enhanced heterogeneous freezing and
prolonged ice crystal growth at higher INP loading.

This competition between homogeneous and heterogeneous freezing has been
discussed in previous studies (Heymsfield et al., 2005; Deng et al., 2018; Takeishi
and Storelvmo, 2018). In contrast, simulations of mixed-phase moderately deep
convective clouds by Miltenberger and Field (2021) indicate that cloud ice mass
concentration increases with increasing INP concentration, which is in opposition to
the findings in this work. The main reason is that the CTT is about -18°C in
Miltenberger and Field (2021)'s study, and heterogeneous freezing does not
compete with homogeneous freezing. Thus, results on INPs effects on glaciation
processes in convective clouds can be opposite under different conditions.

### 3.3.  Cloud liquid mass fraction

Varying the INP concentration has a direct impact on the primary ice formation.
Thus, it affects cloud liquid mass fraction within the clouds (directly for all cloudy
layers where heterogeneous freezing is active and indirectly for warmer and colder
temperatures) and at the cloud top. Cloud liquid mass fraction is defined as the ratio
of mass mixing ratio between cloud droplets ($q_c$) and the sum of cloud droplets and
cloud ice crystals ($q_i$). In-cloud liquid mass fraction, sampled at a time interval of 15
minutes between 9:00 to 19:00 from all cloudy pixels, is shown as scatterplots
versus temperature in Figure 6a-d. The corresponding frequencies of the occurrence
of the temperature/liquid fraction bins are shown in Figure 6e-h. Similar analyses
were made by Hoose et al. (2018), but for idealized simulations of deep convective
clouds. In-cloud liquid mass fractions smaller than 0.5 are quite common already at
temperature just below -3 °C except for the case without rime-splintering process



(A×10$^0$_NSIP). The decrease in INP concentrations has limited effects on the in-
cloud liquid mass fraction (Figure 6c and 6g), while a stronger influence has been
found in the case with enhanced INP concentration (Figure 6d and 6h). The number
of pixels having high liquid mass fraction values at temperatures lower than -30 °C
decreases with increasing INP concentration. In addition, more and more pixels
having liquid mass fraction smaller than 0.5 appear with increasing INP
concentration and the number of pure ice pixels increases with increasing INP
concentration as well. This is because higher INP concentration intensifies the
heterogeneous freezing processes (immersion freezing and deposition ice
nucleation) and invigorates the rime-splintering process as well (will be discussed in
section 3.3). Interestingly, at the lower end of the mixed-phase temperature range (-
38 ~ -28 °C), there are fewer pixels having high liquid mass fraction in the high INP
case, and those remaining are mainly the ones at high vertical velocities (above ~ 10
m/s). This is probably because supercooled droplets are more easily frozen in high
INP cases and stronger updrafts are needed to offset the Wegener-Begeron-
Findeisen process to maintain the supersaturation with respect to water. Switching
off the secondary ice production via rime-splintering process, pixels having a liquid
mass fraction smaller than 0.9 are reduced significantly at temperatures between -
10 °C and 0 °C (Figure 6b and 6f).

At the cloud top (Figure 7), the number of pixels having a liquid mass fraction smaller
than 0.5 increases with increasing INP concentration, which is the same as within
the clouds. "Cloud top" is defined as the height of the uppermost cloud layer (which
has a condensed mass of cloud water plus cloud total cloud ice greater than a
threshold of 1.0×10$^{-5}$ kg kg$^{-1}$) in a pixel column. At the cloud top, the liquid mass
fraction has a more polarized distribution, with either large values or small values,
and intermediate values are less common than within the clouds. This is because the
vertical velocities at the cloud top are significantly smaller compared to that within
the cloud, which leads to a more efficient Wegener-Begeron-Findeisen process at
the cloud top.



### 3.4. Liquid cloud pixel number fraction

Liquid cloud pixel number fractions are calculated differently for model simulations and retrieved cloud products. For simulation results, a cloudy pixel having a cloud liquid mass fraction larger than 0.5 is counted as a liquid pixel, otherwise, it is an ice pixel. Both CLAAS-2 and SEVIRI_ML products and the corresponding retrievals derived from ICON simulations by the satellite forward operators (see section 2.4) provide binary cloud phase information (liquid or ice) only. For these data, the liquid cloud pixel number fraction is calculated as the ratio between the number of liquid cloud pixels and the sum of all cloudy pixels.

Liquid cloud pixel number fractions within clouds and at the cloud top are shown in Figure 8. Decrease in INP concentration has limited impacts on the liquid cloud pixel number fraction for in-cloud layers. Increase in INP concentration leads to a decrease in liquid cloud pixel number fraction but not monotonically (Figure 8a). The decrease in liquid cloud pixel number fraction is significant in the highest INP concentration case (case A$\times10^3$), while decreases in intermediate INP concentration cases (cases A$\times10^1$ and A$\times10^2$) are only obvious in temperature ranges from -30 °C to -20 °C and from -15 °C to -5 °C. Switching off the rime-splintering process results in an increase in liquid cloud pixel number fraction in the temperature range between -10 °C and -3 °C, which is consistent with the strong decrease in pixels of cloud liquid mass fraction lower than 0.9 in the same temperature range (Figure 7b). The temperature at which the liquid cloud pixel number fraction equals 0.5 is often termed "glaciation temperature". The glaciation temperature shifts slightly to a warmer temperature by ~2 °C at the highest INP concentration case (case A$\times10^3$, Figure *8*a).

Sensitivities of the cloud phase to INP concentration are more complex at the cloud top than inside the cloud. Liquid cloud pixel number fractions at the cloud top calculated directly from ICON simulations on its native grid (~1200 m) are shown in Figure 8b. Cloud-top liquid pixel number fraction decreases significantly with increasing INP concentration. In the temperature range between -35 °C and -15 °C, where heterogeneous freezing processes (immersion freezing and deposition nucleation) are dominant, the impact of INP is most pronounced. Above -15 °C, the





impact of INP does not disappear, especially in the highest INP concentration case
(case A$\times 10^3$). This is mostly likely due to the sedimentation of ice crystals from upper
layers and the secondary ice production invigorated by the Wegener-Begeron-
Findeisen process. Switching off the rime-splintering process increases cloud-top
liquid pixel number fraction only slightly in the temperature range from -10 °C to -
3 °C and is almost identical to the control run (case CTRL) outside this temperature
range. Interestingly, the shift of glaciation temperature with increasing INP
concentration is about 8 °C (Figure 8b) at the cloud top, which is stronger than that
inside the clouds (~2 °C, Figure 8a). A possible explanation is that, typically, the
vertical velocity at the cloud top is smaller than within the cloud and the ice formation
through the Wegener-Bergeron-Findeisen process is expected to be more efficient.
Thus, the Wegener-Begeron-Findeisen process is more sensitive to INP perturbation
at the cloud top than within clouds, and leads to the glaciation temperature shifting to
be more significant at the cloud top.

Liquid cloud pixel number fractions at the cloud top calculated directly from ICON
simulations on SEVIRI's grid (~ 5000 m) are shown in Figure 8c. They are noisier
and do not exhibit the small minimum between -10 °C and -3 °C related to rime-
splintering, but are otherwise very similar to Figure 8b. In contrast, the scale-aware
and definition-aware ICON_RTTOV_CLAAS-2 cloud-top liquid pixel number fractions
shown in Figure 8d differ markedly from the direct or regridded model output. Above
-23 °C, increase and decrease in INP concentration both lead to a decrease in cloud-
top liquid pixel number fraction at certain temperature, but the high INP
concentration cases (cases A$\times 10^2$ and A$\times 10^3$), still exhibit the lowest liquid fractions,
and case A$\times 10^0$_NSIP the highest. Thus, the fingerprints of primary and secondary
ice formation are retained in the ICON_RTTOV_CLAAS-2 liquid fraction in this
temperature range only for very strong perturbations. At the same time, it must be
noted that the decrease of the liquid pixel number fraction to values around 0.8
above -15 °C is not related to the rime-splintering process, but to the application of
the CLAAS-2 satellite simulator.

Below -23 °C, in the high INP cases A$\times 10^2$ and A$\times 10^3$, cloud-top liquid pixel number
fractions even increase with increasing INP concentration. In moderate and low INP





cases, the impacts of INP perturbation are not pronounced. Moreover, the shape of
cloud-top liquid pixel number fraction decreasing with cloud-top temperature is
different from that in Figure 8b. Here, the fingerprints of the ice formation processes
are completely lost. As demonstrated in Figure 8c, remapping of simulation data onto
SEVIRI's coarser grid is not the cause of liquid pixel number fraction difference
between direct ICON output and the ICON_RTTOV_CLAAS-2 diagnostics, but the
loss of information through the postprocessing is responsible.

The satellite observed cloud-top liquid pixel number fraction from CLAAS-2 is plotted
as a grey dashed line in Figure 8d. It does not reach 1.0 for all cases even as the
cloud-top temperature is approaching 0 °C, and shows a different temperature
dependency than the simulated curves. No matter how strong the INP concentration
and rime-splintering are perturbed, the retrieved cloud-top liquid pixel number
fractions from simulation data deviate strongly from the CLAAS-2 products. In this
context one should note that in particular cloud edges have been found to be
problematic situations for the cloud retrievals, being to some extent responsible for
biasing the liquid-pixel fraction towards smaller values, in particular for the CLAAS-2
data.

Finally, the comparison to observations is repeated with the SEVIRI_ML retrieval
scheme applied to both simulated radiances (ICON_RTTOV_SEVIRI_ML) and the
SEVIRI observations themselves (Figure 8e). As SEVIRI_ML provides uncertainty
estimates, pixels for which either the cloud mask uncertainty or the cloud phase
uncertainty is larger than 10% are filtered out. While this ensures that only very
certain values are kept, it has a significant impact on the number of remaining values
as more than 90% of the pixels are filtered out. The resulting liquid pixel number
fractions ICON_RTTOV_SEVIRI_ML bear a much stronger similarity to the regridded
model output in Figure 8c. Remaining differences are a noiser behavior, a plateau of
non-zero liquid pixel number fractions even below -40 °C, and a general shift to
lower temperatures. SEVIRI_ML applied to observations (dashed black line in Figure
8e), with the same uncertainty criterion, exhibits the expected behavior with a liquid
fraction of approximately 1 above -10 and 0 °C below approximately -30 °C, and
results in a very good agreement to the $A \times 10^3$ case.



**3.5. Sensitivity of cloud phase to atmospheric stability perturbations**

In addition to the reference run (case CTRL), four cases with perturbations in initial temperatures are analyzed. Mean updraft velocities increase gradually from the low CAPE case INC05 to high CAPE case DEC05 (Figure 9) and cause differences in cloud microphysics and cloud phase distributions.

In-cloud and cloud-top liquid cloud pixel number fractions for the five cases are shown in Figure 10. Systematic shifting of liquid cloud pixel number fractions is detected both inside clouds and at the cloud top. Liquid cloud pixel number fraction decreases with increasing CAPE from INC05 to DEC05. Both in-cloud and cloud-top glaciation temperatures shift toward warmer temperatures as the CAPE increasing from case INC05 to DEC05. This is different from the results reported by Hoose et al. (2018) that cloud-top glaciation temperatures hardly changed with increasing temperature in the boundary-layer by 2 °C, and appears to be contradictory to the expectation that stronger vertical velocities result in a lower glaciation temperature due to suppression of the Wegener-Bergeron-Findeisen process (Korolev, 2007). Further analysis (not shown) revealed that the mass concentration of cloud ice particle increases while the mass concentration of cloud droplet decreases with the increase in CAPE from case INC05 to DEC05. Moreover, homogeneous and heterogeneous freezing are both enhanced in the high CAPE cases (Figure *11*), possibly due to more transport of moisture to upper levels in the stronger updrafts (Figure *9*). With more ice generated, the Wegener-Begeron-Findeisen process can be stimulated despite the higher updrafts. Interestingly, cloud-top liquid pixel number fractions from the two high CAPE cases (cases DEC03 and DEC05) are closer to SEVIRI observations, both using the CLAAS-2 retrieval (Figure 10c) and the SEVIRI_ML retrieval (Figure 10d), especially in the temperature range between -10 and -28 °C. Overall, perturbing initial thermodynamic states or CAPE of convective clouds is as important as the modifications to cloud heterogeneous freezing parameterizations.

**4. Conclusions**

Remote sensing products, which cover the entire globe, provide a unique opportunity to constrain the representation of cloud microphysics in global and regional



numerical models. In this study, instead of comparing simulation results to satellite
observations directly, we derived cloud properties using a radiative transfer model
and two different satellite remote sensing retrieval algorithms and then performed the
comparison. This enables us to evaluate cloud microphysical processes of numerical
models using satellite observations directly. A series of numerical experiments were
performed applying convection-permitting simulations with perturbations in INP
concentrations and initial thermodynamic states to investigate their impacts on cloud
phase distributions in deep convective clouds. Simulation results were compared to
cloud properties derived from SEVIRI measurements to evaluate the model
performance in simulating cloud-top microphysical properties.

INP concentration was found to have a significant role in shaping cloud phase
distributions both within clouds and at the cloud top. Cloud liquid pixel number
fraction decreases with increasing INP concentration both within the cloud and at the
cloud top, indicating a higher glaciation temperature and more intense
heterogeneous freezing processes in enhanced INP concentration cases.
Interestingly, the influences of INP do not increase linearly but are more pronounced
in the high INP concentration cases. In addition, the shifting of glaciation temperature
is more significant at the cloud top than within the cloud, which means the impact of
INP concentration on cloud phase distribution is more pronounced at the cloud top.
This has implications for analyzing cloud products retrieved from passive remote
sensing observations. It turned out that with the CLAAS-2 retrieval scheme, the INP
sensitivity of the cloud-top phase distribution was not detectable, while the
SEVIRI_ML retrieval scheme, for which the most uncertain pixels could be excluded,
resulted in a better agreement and retained the sensitivity to INP. In contrast,
secondary ice production via rime-splintering did not have a detectable impact on the
cloud-top phase distribution. Therefore, in future studies, we recommend using the
SEVIRI_ML retrieval scheme and SEVIRI_ML satellite-based cloud products.

Total cloud ice mass concentrations do not increase but decrease with increasing
INP concentrations in the simulated deep convective clouds. Process rate analyses
reveal that heterogeneous freezing process rates increase with increasing INP
concentrations, while homogeneous freezing process rates decrease with increasing
INP concerns. The competition between heterogeneous freezing and homogeneous



freezing for water vapor suppresses ice formation via homogeneous freezing, which
is the dominant nucleation process in the simulated deep convective clouds, and
hence decreases the cloud ice mass concentration. The increase in heterogeneous
nucleation in high INP cases invigorates riming and collection processes of ice
particles, making it easier for small ice crystals to grow into large ice aggregates and
sediment to lower levels. This is the reason why precipitation increases in enhanced
INP cases.

Perturbations in initial thermodynamic states have a strong impact on the cloud
phase distribution both within the cloud and at the cloud top, although the used
perturbations might be rather large compared to initial condition uncertainty in a
weather forecasting context. To completely distinguish microphysical impacts from
thermodynamic impacts, applying a piggybacking approach (Grabowski, 2015;
Thomas et al., 2023) in future simulations is necessary.

Utilizing satellite forward operator (the RTTOV radiative model) and remote sensing
retrieval algorithms enable us to derive cloud-top microphysical products and
compare simulation results to satellite products more consistently. However, there
are significant differences in retrieved cloud-top liquid fractions between model
simulations and satellite products. The sources of errors are very complicated and
may come from simulation results, satellite operators, or retrieval algorithms, which
will be investigated in the future. Moreover, the cloud-top property analysis
presented in this study is based on domain-wide statistics, including clouds of
varying types. Statistical results could differ if individual clouds are tracked, as clouds
differ in different experiments in terms of locations and extensions. Although there
are significant uncertainties in satellite forward operators and retrieval algorithms,
passively remote-sensed cloud products provide potential opportunities to constrain
microphysical processes in numerical models.

Simulation results of this study reveal a close dependence of heterogeneous
freezing and cloud phase distribution on INP concentrations. Despite this finding, the
ice formation processes in deep convective clouds remain poorly understood. It is
necessary to investigate how and in which conditions the competition of
heterogeneous with homogeneous freezing for water vapor and cloud water depends



on INP availability and vertical velocities in different types of deep convective clouds.
Moreover, the importance of other secondary ice production processes than rime-
splintering (droplet shattering and collisional breakup) in deep convective clouds
need to be quantified in the future.

**Competing interests**
One of the (co-)authors (Corinna Hoose) is a member of the editorial board of
Atmospheric Chemistry and Physics.

**Acknowledgments**
This project has received funding from the European Research Council (ERC) under
the European Union's Horizon 2020 research and innovation programme under grant
agreement 714062 (ERC Starting Grant "C2Phase"). We gratefully acknowledge the
computing time allowed by the German Climate Computing Centre (DKRZ) on the
HPC system Mistral and the Steinbuch Centre for Computing (SCC) on the HPC
system ForHLR II. The contribution of Martin Stengel was supported by EUMETSAT
and its member states through CM SAF.

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



**Tables:**

Table 1: Setups of simulations performed in this study.

| Num | Experiment | Description |
|---|---|---|
| 1 | $A \times 10^0$ (CTRL) | Without any perturbations, the CTRL run, used as a reference. |
| 2 | $A \times 10^{-2}$ | INP concentrations for both immersion and deposition mode are scaled by multiplying parameter A in Equation (1) by $10^{-2}$. |
| 3 | $A \times 10^{-1}$ | Same as num. 2, but multiplying by $10^{-1}$. |
| 4 | $A \times 10^1$ | Same as num. 2, but multiplying by $10^1$. |
| 5 | $A \times 10^2$ | Same as num. 2, but multiplying by $10^2$. |
| 6 | $A \times 10^3$ | Same as num. 2, but multiplying by $10^3$. |
| 7 | $A \times 10^0\_NSIP$ | INP concentration as in CTRL. The secondary ice production (rime-splintering process) is switched off. |
| 8 | DEC05 | Initial and lateral temperature decreases from 3 to 12 km with a maximum increment of 5 K. No perturbations in INPs ($A \times 10^0$). |
| 9 | DEC03 | Same as num. 8, but with a maximum increment of 3 K. |
| 10 | INC03 | Initial and lateral temperature increases from 3 to 12 km with a maximum increment of 3 K. No perturbations in INPs ($A \times 10^0$). |
| 11 | INC05 | Same as num. 10, but with a maximum increment of 5 K. |







**Figures:**


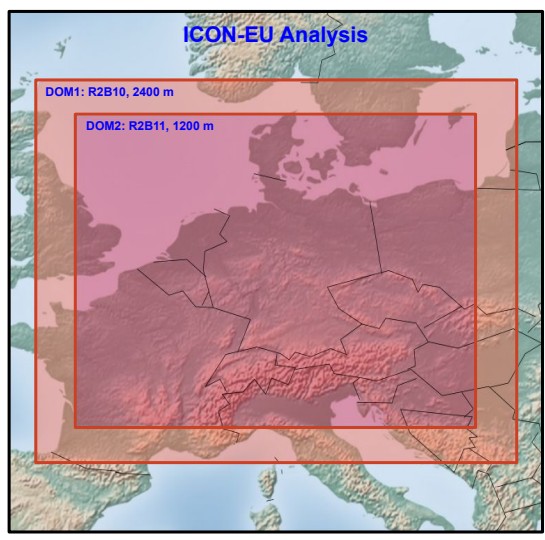


Figure 1: The simulation domains.




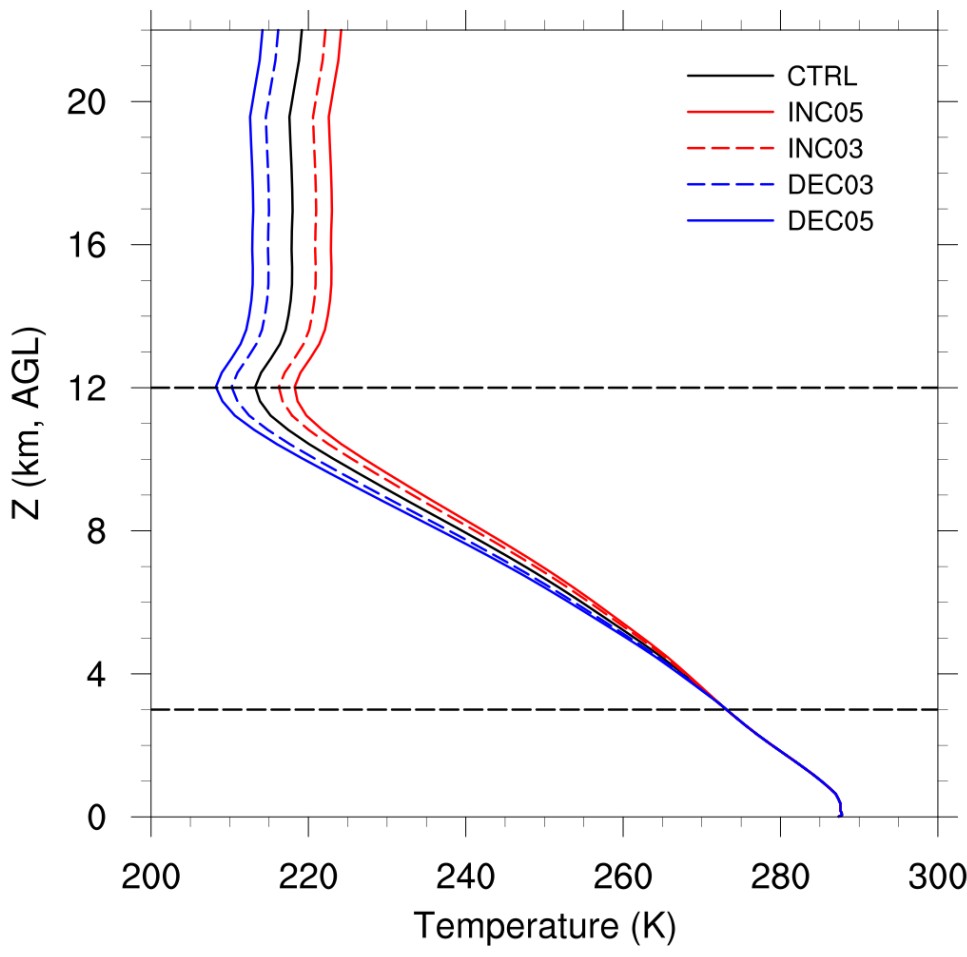


Figure 2: Domain averaged initial temperature profiles. The same modification was
applied to the lateral boundary conditions.

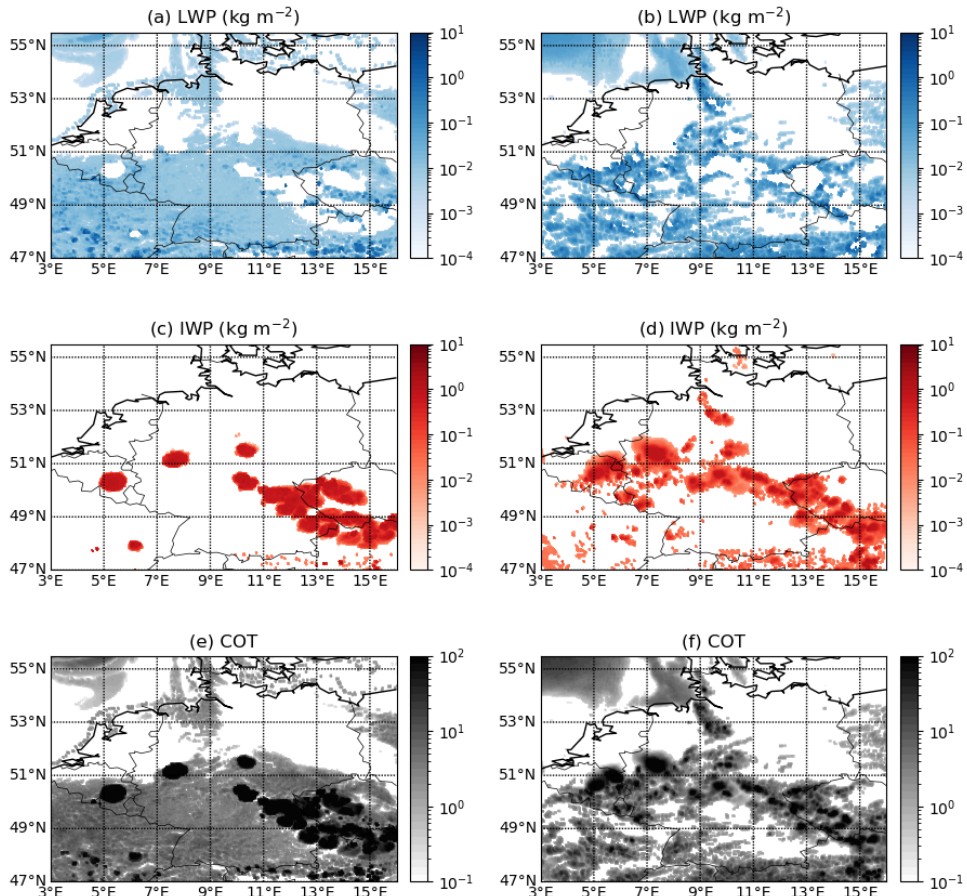

Figure 3: Spatial distributions of retrieved cloud liquid water path (LWP), ice water path (IWP), and cloud optical thickness (COT) at 13:00 UTC. The left panel is for the CTRL case (a, c, e) and the right panel is for the CLAAS-2 product (b, d, f).



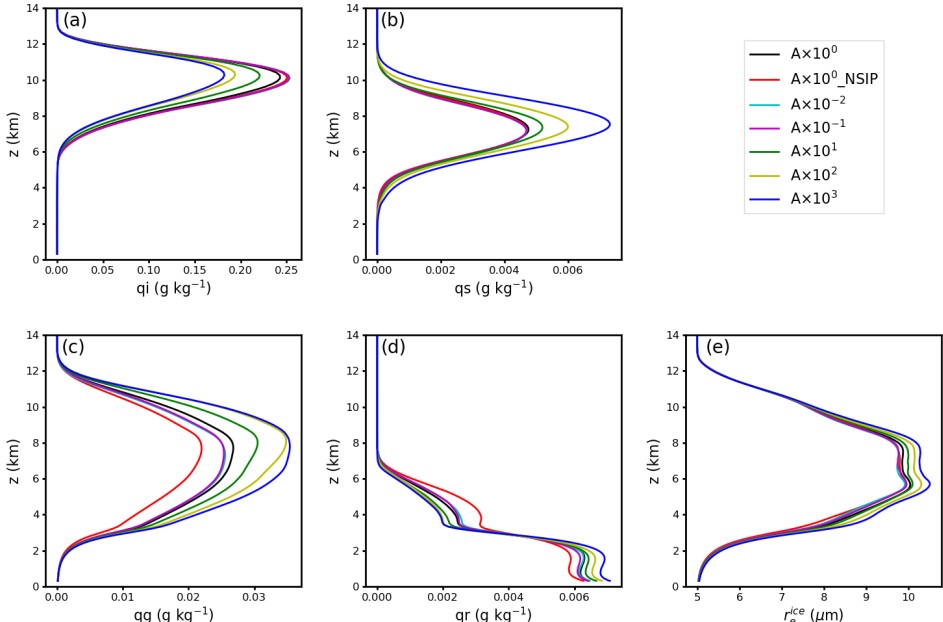

Figure 4: Spatial- and time-averaged (9:00~19:00) profiles of cloud mass mixing
ratios of (a) ice crystals, (b) snow, (c) graupel, (d) rainwater, and (e) ice crystal
effective radius. Mass mixing ratio unit is g kg$^{-1}$ and the unit of ice crystal effective
radius is μm.



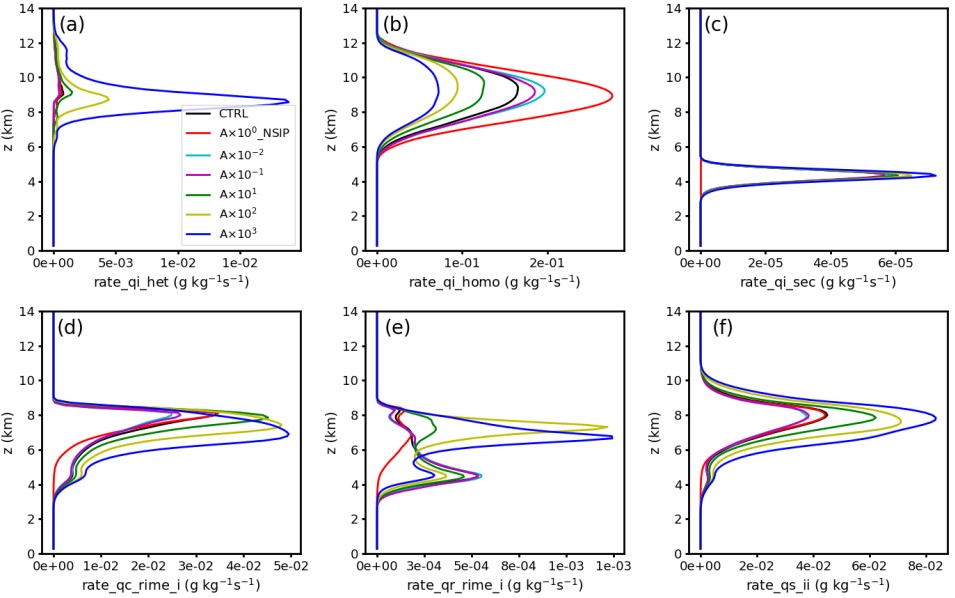

Figure 5: Spatial- and time-averaged (9:00~19:00) profiles of process rates of (a)
heterogeneous freezing (immersion and deposition nucleation), (b) homogeneous
freezing, (c) secondary-ice production (rime-splintering), (d) cloud droplets rimed
with ice crystals, (e) rain droplets rimed with ice crystals, (f) collection between ice
and ice. Unit is g kg-1 s-1. The average mixed-phase layer (0~-38 °C) is roughly in
between 3.2 and 8.6 km. Unit is g kg$^{-1}$s$^{-1}$.

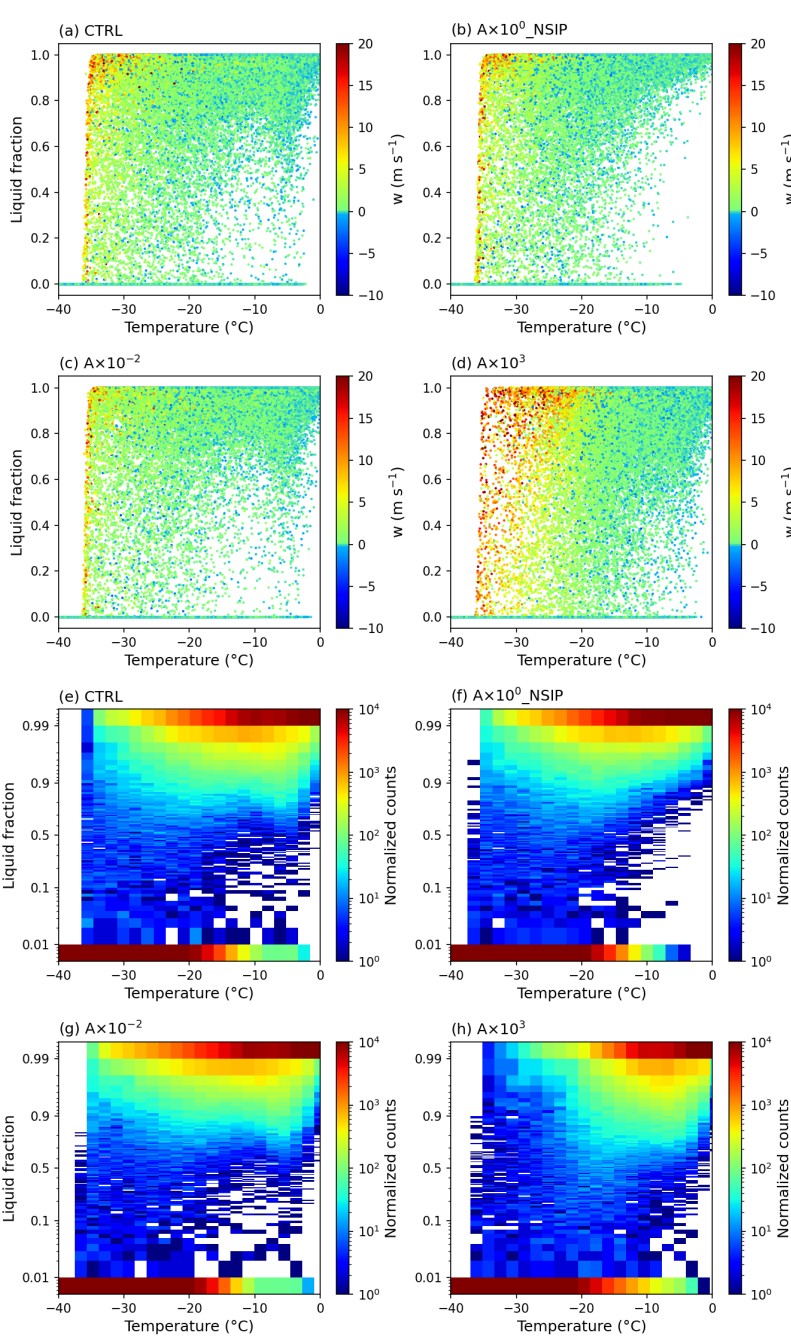


Figure 6: In-cloud supercooled liquid mass fraction distribution as a function of
temperature (binned by 1°C) between 9:00 and 19:00 (a-d) for the 4 cases (A×$10^0$,
A×$10^0$_NSIP, A×$10^{-2}$, A×$10^3$), the colour of points indicates the vertical wind velocity
(unit, m s$^{-1}$). 2-D histogram of in-cloud liquid mass fraction versus temperature (e-f).

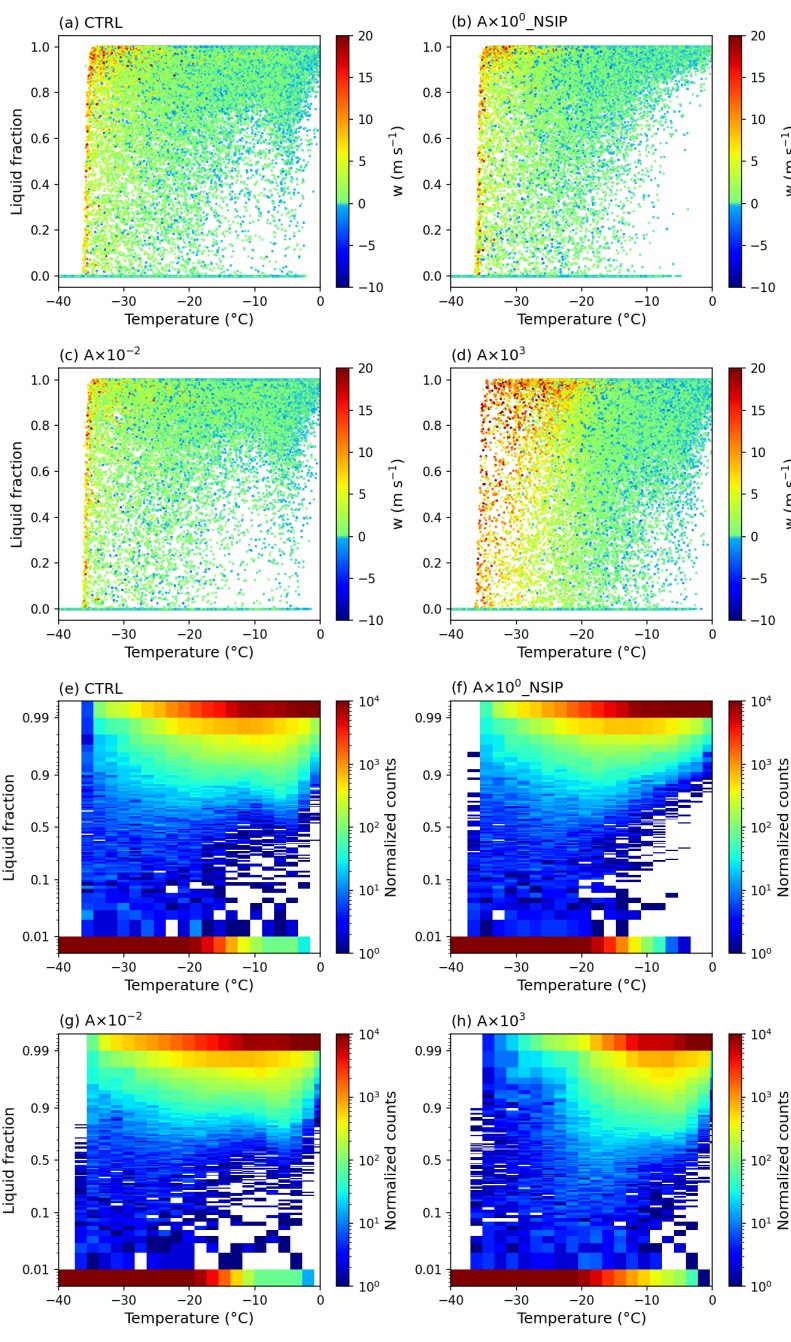

974

Figure 7: Cloud-top supercooled liquid mass fraction distribution as a function of temperature (binned by 1°C) between 9:00 and 19:00 (a-d) for the 4 cases (A×10$^0$, A×10$^0$_NSIP, A×10$^{-2}$, A×10$^3$), the colour of points indicates the vertical wind velocity (unit, m s$^{-1}$). 2-D histogram of cloud-top liquid mass fraction versus temperature (e-f).



979

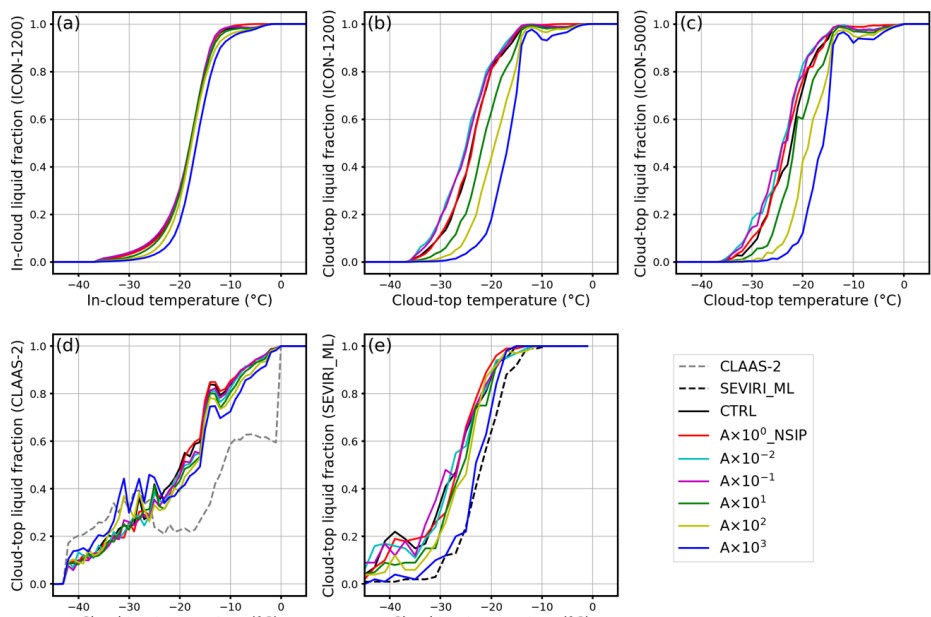

980

Figure 8: Liquid cloud pixel number fraction as a function of temperature from 9:00 to 19:00 UTC for the INP sensitivity experiments, (a) in-cloud fraction calculated from simulations on ICON native grid (~1200 m), (b) cloud-top fraction calculated from simulations on ICON native grid (~1200 m), (c) cloud-top fraction calculated from simulations on SEVIRI's grid (~5000 m), (d) cloud-top fraction calculated by remote-sensing retrieval algorithms to produce CLAAS-2 dataset, and (e) cloud-top fraction calculated by remote-sensing retrieval software suite SEVIRI_ML. The temperature is binned by 1°C in (a), (b), (c), and (d), and by 2°C in (e).





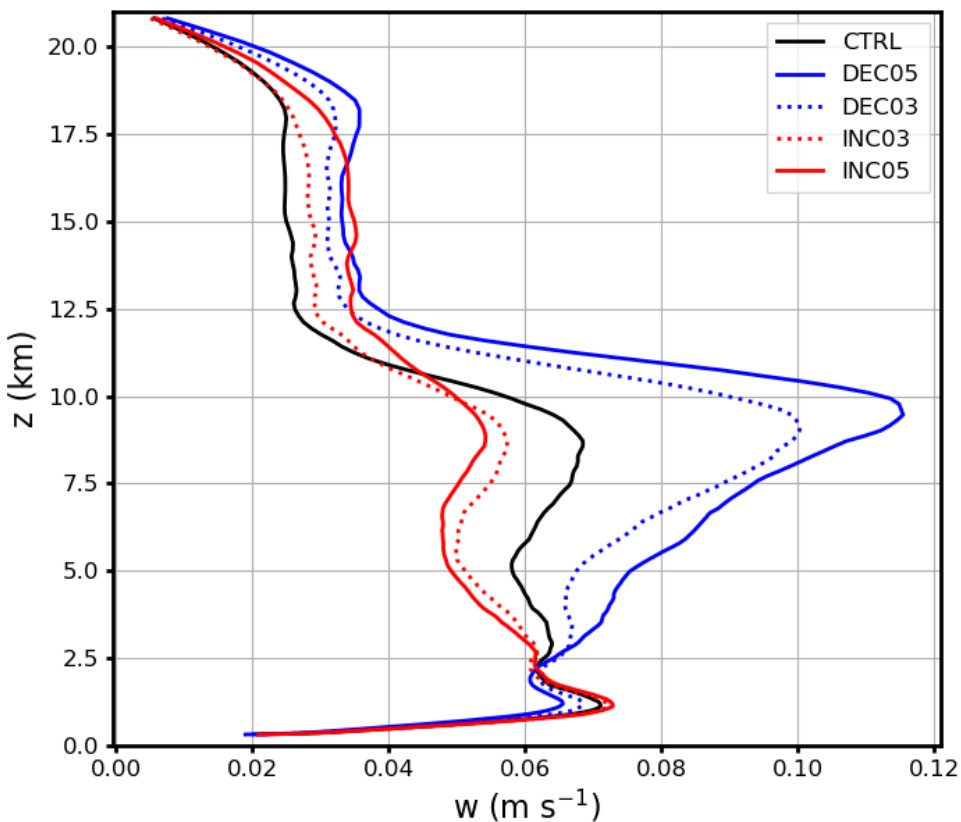


Figure 9: Spatial- and time-averaged (9:00~19:00) profiles of vertical velocities (w
values ≤ 0 m s$^{-1}$ are excluded).




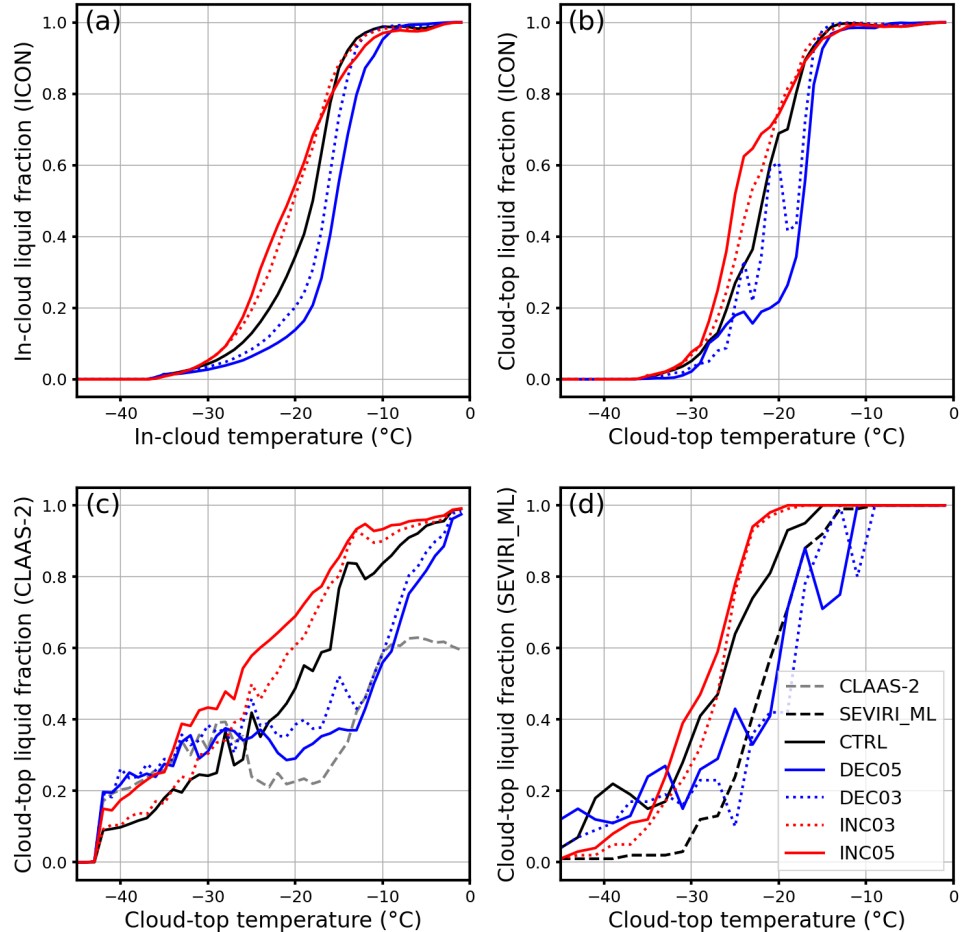


Figure 10: Liquid cloud pixel number fraction as a function of temperature from 9:00 to 19:00 for the thermodynamic sensitivity experiments, (a) in-cloud fraction calculated directly from simulations, (b) cloud-top fraction calculated from directly simulations, (c) cloud-top fraction calculated by remote-sensing retrieval algorithms to produce CLAAS-2 dataset, and (d) cloud-top fraction calculated by remote-sensing retrieval software suite SEVIRI_ML. The temperature is binned by 1°C in (a), (b), and (c), and by 2°C in (d).





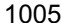

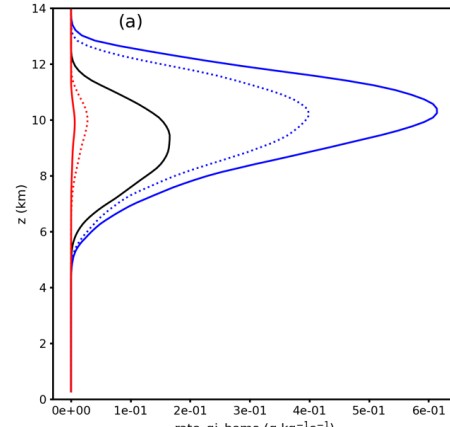 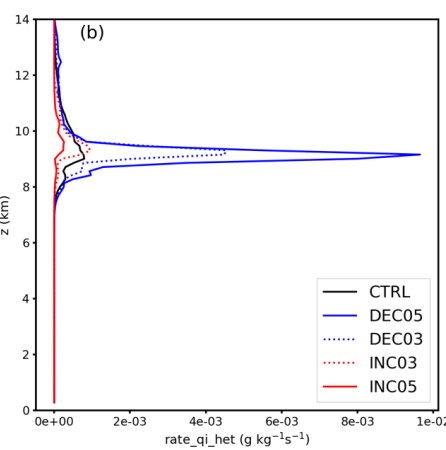


Figure 11: Spatial- and time-averaged (9:00~19:00) profiles of process rates of (a)
homogeneous freezing, (b) heterogeneous freezing (immersion and deposition
nucleation) for cases with perturbed initial thermodynamic states. Unit is g kg$^{-1}$ s$^{-1.}$