# Peer review of "Sensitivity of cloud phase distribution to cloud microphysics and"

_EGUsphere, 2023_

## Author Comment (AC1)

Response to Referee's Comments:

We would like to thank the Editor and the Referee for the time and efforts handling and reviewing our manuscript. The constructive comments and suggestions were very helpful to improve the manuscript.

The Referee's original comments are formatted in black, while our point-by-point responses are formatted in blue font. All the corresponding revisions in the revised manuscript are indicated in red.

**Review #1:**

Han et al. examines the role of ice nucleating particles (INPs) and thermodynamic conditions on mixed-phase microphysics, with particular focus on the proportion of liquid to ice in deep convective clouds. They use cloud-resolving simulations to show that increasing the INP concentration leads to glaciation at warmer temperatures and a lower fraction of liquid cloud pixels, and they examine the microphysical mechanisms driving this effect. Furthermore, they apply multiple satellite forward operators to their model output and compare the results to remotely observed cloud properties in order to explore how well these changes driven by INP concentrations might be detected from satellites.

Overall, I think this study tackles valuable questions given the uncertainty surrounding mixed-phase clouds and their interactions with INPs. The use of satellite forward operators to make an apples-to-apples comparisons of remote sensing observations and model data is novel and interesting. That being said, the paper is not always clear about the causes for discrepancies between the models and observations and how these comparisons contribute to the overall goals and science questions. I have some questions/suggestions on these and a few other points that may improve the analysis and focus the discussion. I therefore recommend this article be accepted with major revisions. Please see more detailed comments below.

We would like to thank the reviewer for the help comments and suggestions and for recognizing the contributions made by this work. Our point-by-point responses are found below.

Major Comments:

1. **Role of comparison with satellite data:** The satellite forward operators allow the authors to directly compare model simulated properties to remote sensing retrievals, but it is not clear whether the goal of these comparisons is to evaluate the performance of the model microphysics or of the forward operators/retrieval algorithms.

   In some places, it seems that the aim is to evaluate how well the model replicates the remotely sensed cloud phase distributions (i.e., the satellite obs are treated as the "ground truth"). In the conclusions, the authors state that the goal is to "evaluate cloud microphysical processes of numerical models using satellite observations directly". However, the discussion does not focus on which processes drive the discrepancies between the modelled and actual cloud phase distributions. If this is the overall goal, then there needs to be more analysis of the microphysical differences between simulations which do match the observed data well (e.g., the DEC03/05 simulations) and those which do not, to actually evaluate which processes are not being treated correctly in the models. In other places, it seems that the aim is to evaluate the retrieval algorithms and whether they are able to capture the differences that we expect to see based on the simulations (i.e., the model data fed through the forward operators is the "ground truth"). Most of the discussion fits this framing, which also makes sense given the comparisons of the CLAAS-2 and ML-based retrievals. The discussion about why the CLAAS-2 retrieval performs worse than the ML retrieval should be expanded on beyond saying it was due to the "loss of information through the postprocessing" (e.g., were the pixels with high uncertainty in the ML retrievals on cloud edges?). Also, why does the CLAAS-2

retrieval not capture the differences in cloud phase distribution as a function of INP when it does capture differences as a function of the thermodynamic environment, despite the two perturbations having a similar order of magnitude effect on the modelled phase distribution at cloud top? Either of the two options could make for a valuable contribution to the literature—but the authors should clarify what they are aiming for in comparing the satellite observations to the models.

Thank you very much for your constructive comments. Our aim is "**to evaluate the retrieval algorithms and whether they are able to capture the differences that we expect to see based on the simulations**" but not to evaluate the model microphysics by comparing with satellite observations.

We revised our manuscript according to your comments and suggestions and clarified our research aims. At the end of "Introduction" section we pointed out our research goal, lines 142 to 144 in the clean version of the revised manuscript "…*We aim to evaluate the satellite retrieval algorithms and investigate whether passive satellite cloud products can detect cloud microphysical and thermodynamical perturbations….*". Moreover, in the "Conclusions" section, we re-emphasized the research goals, lines 609 to 619 in the clean version of the revised manuscript "*……This enables us to make apples-to-apples comparisons between model simulations and satellite observations. ……. Moreover, cloud properties were derived using a satellite forward operator and retrieval algorithms with ICON simulations as input, and compared with CLAAS-2 and SEVIRI_ML satellite cloud products to evaluate whether satellite retrievals could detect perturbations in cloud microphysics and thermodynamics. Uncertainties in the satellite forward operator were however not assessed in this study, which may influence the validity of corresponding results in some extent…….*"

Regarding why SEVIRI_ML performs better than CLAAS-2, we have discussed the reason in the last paragraph of section 3.4 "Liquid cloud pixel fraction" from line 557 to 574 in the clean version of the revised manuscript: "*……As SEVIRI_ML*

*provides uncertainty estimates, pixels for which either the cloud mask uncertainty or the cloud phase uncertainty is larger than 10% are filtered out. While this ensures that only very certain values are kept, it has a significant impact on the number of remaining values as more than 90% of the pixels are filtered out. The resulting liquid pixel fractions ICON_RTTOV_SEVIRI_ML bear a much stronger similarity to the regridded model output in Figure 8c. ……Generally, the SEVIRI_ML retrieval algorithm is assumed to perform better than the CLAAS-2 scheme for both cloud top temperature and cloud phase. This is because SEVIRI_ML employs state-of-the-art neural networks to emulate CALIOP v4 data. Moreover, SEVIRI_ML provides uncertainty estimates which facilitates fliting out pixels with high uncertainties. Nevertheless, retrieval inaccuracies are unavoidable for passive satellite retrievals which holds true for CLAAS-2 but also for SEVIRI_ML……"*

The reviewer also asked why the CLAAS-2 retrieval captures the perturbation in thermodynamics but cannot capture perturbations in INP concentrations at the cloud top. This question also puzzles us. The effect of perturbation in thermodynamics on the cloud phase distribution at the cloud top is as large as the largest INP perturbation (case $A \times 10^3$). However, the impact of thermodynamical perturbation is significantly stronger than the INP perturbation within the cloud (Figure 10a VS Figure 8a in the revised manuscript). Moreover, the impacts of thermodynamical perturbation on domain-averaged profiles of cloud hydrometeors and process rates related to ice cloud process are also significantly stronger than the INP perturbation. Thus, the reason is most likely that the thermodynamical perturbation is stronger than the INP perturbation when the entire depth of the cloud is considered.

2. **Model representativeness:** Section 3.1 compares the retrieved cloud properties between the ICON model using a satellite forward operator and actual satellite observations. There is some discussion of the differences between the model and observations which is generally attributed to "model physics", but it would be useful to see at least a bit more discussion of the factors potentially causing this

(e.g., 1.2km grid spacing isn't resolving entrainment fully). Though I do appreciate that the simulation is not going to perfectly match the observations, the article would be improved if the authors consider how any under-resolved model physics might affect the validity of their findings (or at least argue why they think this isn't the case). For example, a good amount of the discussion focuses on differences in ice microphysics between cloud top and in-cloud—if entrainment is under-resolved, is it possible that the simulated difference between the two regions is magnified compared to reality? The results here may certainly still be applicable especially towards cloud cores that more closely resemble the very homogenous clouds simulated here, but a caveat about realistic cloud edges might improve the discussion.

Thank you very much for your constructive comments and suggestions. We have added more discussions on the potential factors causing the discrepancies between model simulations and satellite observations. More importantly, we have extended the discussion on the potential cloud physical factors, including entrainment mixing process and secondary ice production processes. See the following sentences in the clean version of the revised manuscript from line 343 to 358 "……*The error sources are manifold and may originate from the model physics as well as from the forward operator and the retrieval algorithm. Geiss et al. (2021) investigated the sensitivity of derived visible and infrared observation equivalents to model physics and operator settings. They found that the uncertainty of the visible forward operator is sufficiently low while infrared channels could bring errors in cloud-top variables. Geiss et al. (2021) concluded that the primary source of deviations is mainly from model physics, especially model assumptions on subgrid-scale clouds. In addition to the subgrid-scale cloud scheme, multiple critical cloud microphysical processes missing from the model, introducing significant uncertainties into the simulation results. For example, the entrainment mixing process is not resolved or parameterized in the model, which has essential influences on processes at cloud boundaries and hence the cloud properties (Mellado, 2017). Moreover, secondary ice processes including droplet shattering and collisional breakup due to ice*

*particles collisions are missing, which have significant impacts on cloud ice microphysics (Sullivan et al., 2018; Sotiropoulou et al., 2021)……"*

3. **Figure 6 and Figure 7 are the same figure.** The authors have probably inserted the wrong figure for one of these unless I'm missing something here.

   Thanks for noting this. We inserted a wrong figure, apologies. The correct figure has been inserted for Figure 7 in the revised manuscript.

**Minor Comments:**

1. Lines 86-87: Suggest citing some work on the transition to deep convection around mixed phase regions (e.g., Li et al., 2013, Sheffield et al., 2015, Mecikalski et al., 2016).

   References have been added.

2. Lines 112-123: A lot of this paragraph focuses on INP impacts on precipitation, but this doesn't seem to be a focus of the rest of the article, so this can be abbreviated to a short statement that the impact of INPs on precipitation from deep convective clouds is still uncertain and may depend on precipitation/cloud type.

   The sentences have been revised to "*Both observations and simulations reveal that INPs impact deep convective cloud properties including the persistence of deep convective clouds and precipitation (Twohy, 2015; Fan et al., 2016). However, the impact of INPs on precipitation from deep convective clouds is still uncertain and may depend on precipitation and cloud types (van den Heever et al., 2006; Min et al., 2009; Fan et al., 2010; Li and Min, 2010).*"

3. Lines 141-146: These sentences/references would fit better in the methods.

   The sentences have been moved to the "Satellite observations and retrieval algorithms" section, from line 286 to 288 in the clean version of the revised manuscript.

4. Lines 183-193: Are the fitting constants described uniform in space and time, or does the INP concentration only depend on the ambient temperature?

The fitting parameters in Equation (1) vary seasonally but do not vary in space. Thus, the INP concentration not only depends on the ambient temperature but also has seasonal variations.

5. Lines 204-205: How frequently is the model output/sampled for the analyses?

The output interval is 15 minutes. The sentence "……*Simulation results were saved every 15 minutes……*" has been added in the revised manuscript.

6. Lines 222-225: The description of the temperature perturbations was confusing. Suggest rephrasing to "The temperature increment is linearly increased/decreased with height from 0K at 3km to +/-3 or +/-5K at 12km, […]".

The sentence has been revised to "*……The temperature increment is linearly increased/decreased with height from 0 K at 3 km to +/-3K and +/-5K at 12 km, creating 4 sensitivity experiments DEC03, DEC05, INC03, and INC05……*" in the revised manuscript.

7. Lines 347-351: Is there a portion of the domain near the lateral boundaries that is excluded from the analysis?

Yes, there are 4 cell rows for boundary conditions and a nudge zone, in a total of about 30 pixels at the edge of the inner domain, which has been excluded from the analysis.

8. Section 3.2: Does changing the time period considered impact the results (here and in the rest of the sections) at all? Are the trends as a function of INP concentration relatively consistent in time?

Changing the average time period has some impacts on the results, as the developing stage of convective clouds is changed. The trends and statistics analyzed

in this study are still as a function of INP concentration. As the overall results and conclusions do not change, we stick to the chosen time period.

9. Fig 4, 5, 8: The color scheme used makes it difficult to see trends as a function of INP. Maybe use a color scheme with warmer colors for increased INP and cooler colors for decreased INP (or something similar).

Warmer colors were used for increased INP and cooler colors were used for decreased INP cases in Figs 4, 5, and 8 in the revised manuscript.

10. Fig 4-11: Figures would all be improved by indicating important altitudes/levels with a dashed line or similar, or adding a shaded region to indicate the mixed phase region.

Shaded areas indicating the spatial- and time-averaged mixed phase region have been added in Figs 4, 5, and 11 in the revised manuscript. Figures 6, 7, 8, and 10 present quantities as a function of temperature and the mixed-phase region is directly seen, thus no shaded areas or dashed lines are added in those figures.

11. Fig 5: It would be helpful to add a panel to show what percentage of the overall ice formation is heterogeneous vs. homogeneous.

Ice formation is dominated by the homogeneous freezing in this simulation study. Fig 5a and 5b in the revised manuscript show process rates for heterogeneous freezing and homogeneous freezing, respectively. It is clearly seen that process rates of homogeneous freezing (Fig 5b in the revised manuscript) are larger than the process rates of heterogeneous freezing (Fig 5a in the revised manuscript) by one to two orders of magnitude. Moreover, there are already multiple subplots in Fig 5. Thus, we decided not to add an additional panel in Fig 5 in the revised manuscript to show what percentage of the overall ice formation is due to heterogeneous freezing or homogeneous freezing.

12. Fig 6-7: This figure format could use a dashed line at 0.5 liquid fraction since the

glaciation temperature is referenced multiple times in the discussion. Also, it is hard to see the differences in the panels e-h, a difference plot relative to the control instead (or in addition) would be clearer.

Dashed lines at 0.5 liquid fraction which indicate the glaciation temperature have been added in Figs 6 and 7 in the revised manuscript. As for your second suggestion, it is impossible to make it. When calculating the liquid fraction for each case, the data points were selected randomly and the number of the points is different for each case. In the same temperature bin for different cases, the counts of the data points are different from each other. Thus, it is impossible to simply calculate the differences to the CTRL case. The 2-D histograms shown in panels e-h stand for the density of the data points. When one analyzes them together with the liquid mass fraction distributions (panels a-d), the differences between the four cases are clearly seen. Thus, we think the figures are clear enough as they are and difference plots wouldn't tell more.

13. Section 3.3: Are the trends in liquid mass fraction presented here monotonic/is there a consistent trend among all the INP concentrations tested? The authors don't need to show these in the article itself but would be good to make a statement about that and include those figures as a supplement or as a reply to this comment.

    The liquid mass fractions as a function of temperature within the cloud and at the cloud top are shown in Figure 1. The temperature was binned by 1 ˚C, and the mean value of liquid mass fraction was calculated in each temperature bin. Liquid mass fraction decreases monotonically with increasing INP concentration in the temperature range from about -15 to -35 ˚C both within the cloud and at the cloud top (except for the lowest INP concentrations), and the decreasing trend is more significant at the cloud top compared to within the cloud.

[Figure]

Figure 1: Liquid mass fraction as a function of temperature from 9:00 to 19:00 UTC for the INP sensitivity experiments, (a) in-cloud liquid mass fraction, (b) cloud-top liquid mass fraction. The temperature is binned by 1 ˚C.

14. Section 3.4: "Liquid cloud pixel number fraction" is a confusing term. Especially since Section 3.3 is about the "cloud liquid mass fraction", I initially thought "number fraction" was referring to the number of liquid drops versus ice crystals, rather than the number of pixels which are mostly liquid. Maybe the term could be changed to something like "liquid cloud pixel fraction"?

"Liquid cloud pixel number fraction" has been changed to "liquid cloud pixel fraction".

15. Lines 605-606: The authors say that the magnified impact of INPs at cloud top compared to in-cloud has "implications for analyzing cloud products […]". What are the implications?

The sentence is confusing and has been deleted in the revised manuscript.

16. Lines 614-615: Found this sentence confusing: "Total cloud ice mass concentrations do not increase but decrease with increasing concentrations in the simulated deep convective clouds." Does this just mean that the total cloud ice mass decreases with increasing INP concentrations?

As indicated in Figure 4a, the spatial- and time-averaged profiles of mass concentration of cloud ice crystals decrease with increasing INP concentrations.

The sentence has been revised to "*Ice crystal mass concentration does not increase but decreases with increasing INP concentrations in the simulated deep convective clouds.*" to make it clearer.

17. Lines 627-632: Add a sentence here about the relative impacts of INP and thermodynamic perturbations on the cloud phase distribution.

The sentence "……*Moreover, cloud thermodynamics can perturb the cloud phase distribution even stronger than microphysics……*" has been added in lines 654 to 655 in the clean version of the revised manuscript.

---

## Author Comment (AC2)

Response to Referee's Comments:

We would like to thank the Editor and the Referee for the time and efforts handling and reviewing our manuscript. The constructive comments and suggestions were very helpful to improve the manuscript.

The Referee's original comments are formatted in black, while our point-by-point responses are formatted in **blue** font. All the corresponding revisions in the revised manuscript are indicated in **red**.

**Reviewer #2:**

Within the manuscript, the authors retrieve cloud properties from ICON simulations of a deep convective cloud day over central Europe using a satellite forward operator and remote sensing retrieval algorithms. They test the influence of lower and higher concentrations of ice-nucleating particles (INPs) on the liquid-to-ice partitioning in-cloud and at cloud top as well as how different initial thermodynamic states influence the cloud microphysics. The authors convincingly show that microphysical and thermodynamic adjustments to the model setup can be equally important for the simulated cloud microphysics. Overall, the high INP and high convective available potential energy (CAPE) scenarios show the best match with satellite observations from SEVIRI.

The manuscript reads exceptionally well with an easy-to-follow structure. From a methodological side, particular emphasize is given to the use of a satellite forward operator for a more 'apples-to-apples' comparison between the ICON model output and the satellite observations. In remapping the ICON output to the SEVIRI resolution before applying the satellite operator, the authors can disentangle the effects of remapping and of the satellite operator on the simulated cloud microphysics.

I want to emphasize that it has been a great pleasure to read the manuscript, and I

recommend it for publication after only very minor revisions. I structured my review into general and specific comments below, before I will give some hints for technical corrections.

We would like to thank the reviewer for the helpful comments and suggestions and for recognizing the contributions made by this work.

General Comments

(1) It looks to me that Figures 6 and 7 are identical and the authors may have accidentally placed the same figure twice into the manuscript? I cannot identify any difference within the two figures. However, based on the text, it sounds like this is an important figure with a very interesting interpretation, so I suggest to double-check Fig. 7 if it indeed shows cloud top.

Unfortunately, we inserted a wrong figure. In the revised manuscript the correct figure has been inserted for Fig. 7, which shows the supercooled liquid mass fraction at the cloud top. The text is correct and describes cloud liquid mass fraction at the cloud top. Sorry for the mistake.

(2) In Figure 3 you compare the spatial distributions of liquid water path, ice water path, and cloud optical thickness for the CTRL case and the CLAAS-2 satellite product. It shows clear discrepancies between the ICON output and the satellite observations in terms of intensity and spatial coverage. The authors state that, as Geiss et al. (2021) reported, the primary source of these deviations stem mainly from model assumptions on subgrid scale clouds. However, as you find a much better match between the model and satellite observations for the high INP case (with SEVIRI_ML), it would be interesting to reproduce Figure 3 for the high INP case and compare it to the satellite observations and see if you find a better match with the LWP/IWP/COT maps.

Thank you very much for your constructive comments. The SEVIRI_ML retrieval software suite is however limited to neural networks for cloud mask, cloud phase, cloud top temperature, cloud top pressure, and cloud base height. A manuscript on the

SEVIRI_ML is under preparing and the code is available on github: https://github.com/danielphilipp/seviri_ml. Thus, the SEVIRI_ML does not provide any LWP/IWP/COT products. Therefore, figures of LWP/IWP/COT can unfortunately not be produced for SEVIRI_ML.

Specific Comments

Line 102: I cannot completely follow the line of argumentation here. With respect to which parameter does the frequency distribution of ice water fraction have a U-shape? Please clarify this.

Using aircraft observations, the ice water fraction of mixed-phase clouds was analyzed in the work by Korolev et al. (2003). They found that the ice water fraction (IWF) has a minimum in the range $0.1 < IWF < 0.9$, and two maxima for $IWF < 0.1$ and $IWF > 0.9$. Thus, the probability distributions of IWF have a U-shape, with two maxima at the two ends and minimum in the middle, which is shown in Figure 5a in the paper of Korolev et al. (2003). To clarify, we revised the sentence to "......*Aircraft-based observations of mixed-phase clouds properties reveal that the frequency distribution of the ice water fraction has a U-shape with two explicit maxima, one for ice water fraction smaller than 0.1 and the other for ice water fraction larger than 0.9, and the frequency of occurrence of mixed-phase clouds is approximately constant when the ice water fraction is in the range between 0.2 and 0.5......*" in lines 104 to 109 the revised manuscript.

Korolev, A., McFarquhar, G., Field, P. R., Franklin, C., Lawson, P., Wang, Z., Williams, E., Abel, S. J., Axisa, D., Borrmann, S., Crosier, J., Fugal, J., Krämer, M., Lohmann, U., Schlenczek, O., Schnaiter, M., Wendisch, M. (2017). Mixed-phase clouds: Progress and challenges. Meteorological Monographs, 58: 5.1-5.50.

Figure 6: As the color bars are the same for Figs. 6a-d and 6e-h, respectively, I suggest to just show one color bar each and rather have a y-axis title at each individual panel, as right now it is a bit confusing with w (m/s) corresponding to the color bar but not to

the y-axis title for the panels on the right column. In addition, could you explain what you mean with normalized counts in panels 6e to 6h? The normalization is a bit unclear to me.

Color bars have been changed in Figs 6 and 7 in the revised manuscript. Subplots a-d shared a color bar and subplots e-f shared the other color bar. Panels e-f in Figs 6 and 7 were plotted using Python's library *matplotlib.pyplot.hist2d*. In order to better present the data, the normalization method is used to scale scalar data to the $[10^0, 10^4]$ range before mapping to colors using color map. The scaling factor for each subplot is different and depends on the highest count in each subplot. Thus, the counts shown in the plots are not the real numbers of data points but is the "Normalized Counts".

L489: Could you discuss at this point how much higher the cloud is extending above the mixed-phase temperature range and can you somehow diagnose the sedimentation rate in the model to investigate this statement?

The deep convective clouds simulated in this study have reached the homogeneous freezing temperature. As indicated in Figure 1 below, the cloud-top temperature of convective cores at the mature stage is lower than -65 ˚C and is far beyond the mixed-phase temperature range. Therefore, there are sufficient ice crystals formed via the homogeneous freezing process. Vertical velocity close to the cloud top is smaller than within the cloud that sedimentation of large ice crystals and the Wegener-Bergeron-Findeisen process are expected to be more efficient. Unfortunately, the sedimentation rates were not stored for the simulations and cannot be diagnosed in hindsight.

[Figure]

Figure 1:Spatial distribution of retrieved cloud-top temperature at 13:00 UTC for the CTRL case (upper) and for the CLAAS-2 product (lower).

L547 and Fig. 8: This is only a suggestion, but as you talk about a temperature shift as compared to Fig. 8c (on the SEVIRI grid), maybe you could move panel e below panel c and move the legend to where panel 8e has been before? Thus, the temperature shift would be immediately clear, and it is a bit easier to compare the shape of the curves.
Fig.8 has been replotted according to your suggestion in the revised manuscript.

Fig. 9: maybe adding a rough estimate where the cloud top height is in these simulations would help to interpret the vertical profiles of vertical velocities.
The cloud top heights are similar in these simulations and a dashed line indicating the

cloud top height has been added in Fig 9 in the revised manuscript.

Technical Corrections

L349: cloud water plus cloud ice

Corrected

L434: that is Sect. 3.3, is this the correct reference?

It was a typo and has been changed to section 3.4.

L547: noisier

Corrected

L551: of approximately 1 above -10°C and 0 below approximately -30 °C,

Corrected

L563: as the CAPE increases

Corrected

---

## Author Response (AR2)

Response to Referee's Comments:

We would like to thank the Editor and the Referee for the time and efforts handling and reviewing our manuscript. The constructive comments and suggestions were very helpful to improve the manuscript.

The Referee's original comments are formatted in black, while our point-by-point responses are formatted in **blue** font. All the corresponding revisions in the revised manuscript are indicated in **red**.

Comments Referee #1

I appreciate the authors' efforts in addressing the reviewer comments/suggestions. The manuscript is now greatly improved, and I have just a few minor comments before I think this paper can be published.

Thank you very much for your helpful comments and suggestions.

1. I appreciate the clarification of the manuscript's aim of evaluating satellite retrieval algorithms in the introduction and conclusion sections. This aim should be made clearer in the abstract as well.

The sentence "*……Moreover, passive satellite retrieval algorithms and cloud products were evaluated to identify whether they can detect cloud microphysical and thermodynamical perturbations……*" has been added in line 28 to 30 in the clean version of the revised manuscript to emphasize our aim in the abstract.

2. Regarding the comparison between SEVIRI_ML and CLAAS-2, I understand that SEVIRI_ML performs better because it can mask out points with high uncertainty, and this point has been made clearer in the revised manuscript. Is it possible to say anything about which pixels are being filtered out/tend to have higher uncertainty? For example, are they the pixels occurring close to cloud top/edge, or pixels around the mixed phase

region, etc.? This would be a more valuable and more broadly applicable insight.

We carried out a corresponding analysis with the result that filtering of pixels with high uncertainty is spatially rather randomly, thus unfortunately we could not find any pattern that would allow a general statement in this regard. We have included the following sentence in line 561 to 563 in clean version of the revised manuscript: "*......more than 90% of the pixels are filtered out. The filtering affects pixels rather randomly, thus we could not identify any patterns of pixels, such as cloud edges, that are primarily affected by the filtering. ....*"

3. Authors should include the discussion of perturbation in thermodynamics vs. INP concentrations (they can paraphrase from the last paragraph of their response to my first major comment) in the text. The end of section 3.5 would be an appropriate place to add this.

At the end of section 3.5, a paragraph has been added to compare the impacts between perturbation in thermodynamics and INP concentrations. Line 604 to 612 in the clean version of the revised manuscript: "*......Compared to the INP perturbation, the impact of thermodynamical perturbation on cloud phase distribution is significantly stronger within the cloud (Figure 8a and Figure 10a). At the cloud top, the effect of perturbation in thermodynamics on the cloud phase distribution is as large as the largest INP perturbation (case A×103). Moreover, the impacts of thermodynamical perturbation on domain-averaged profiles of cloud hydrometeors and process rates related to the ice cloud process are also significantly stronger than the INP perturbation. Thus, the thermodynamical perturbation is stronger than the INP perturbation when the entire depth of the cloud is considered. Overall, perturbing initial thermodynamic states or CAPE of convective clouds is as important as and may even stronger than the modifications to cloud heterogeneous freezing parameterizations ......*"